# Evaluation of the Change in Density with the Diameter and Thermal Analysis of the Seven-Islands-Sedge Fiber

**DOI:** 10.3390/polym14173687

**Published:** 2022-09-05

**Authors:** Lucas de Mendonça Neuba, Raí Felipe Pereira Junio, Andressa Teixeira Souza, Matheus Pereira Ribeiro, Pedro Henrique Poubel Mendonça da Silveira, Thuane Teixeira da Silva, Artur Camposo Pereira, Sergio Neves Monteiro

**Affiliations:** Department of Materials Science, Military Institute of Engineering-IME, Praça General Tíburcio, 80, Urca 222290-270, RJ, Brazil

**Keywords:** seven-islands-sedge fiber, density, thermal analysis, morphology characterization

## Abstract

Basic properties of sedge fibers from the seven-islands-sedge plant (*Cyperus malaccensis*) were investigated with possible application in reinforcing composite materials. A dimensional distribution and the effect of fiber diameter on density were investigated using gas pycnometry. The Weibull method, used to statistically analyze the acquired data from the diameter intervals, indicated an inverse dependence, where the thinnest fibers had the highest density values. The morphology of the fibers was obtained through scanning electron microscopy (SEM), in which a lower presence of defects was revealed in the thinner fibers, corroborating the inverse density dependence. In addition, the sedge fiber was characterized by differential scanning calorimetry and thermogravimetric analysis, which indicate an initial thermal degradation at around 241 °C. These results revealed for the first time that thinner sedge fibers might be promising reinforcement for polymer composites with a limit in temperature application.

## 1. Introduction

Due to the emerging concern about environmental changes, the possibility of replacing synthetic materials with eco-friendly natural materials has emerged, owing to several advantages; most important, they do not emit greenhouse gases, and are renewable and biodegradable [1,2,3,4]. For this reason, natural lignocellulosic fibers (NLFs) have been widely investigated in recent decades as a possible replacement for synthetic fibers that are traditionally applied as a reinforcing phase in polymer matrix composites [5,6,7]. In spite of their advantages, NLFs also have shortcomings related to their hydrophilic behavior, which hinders adhesion with hydrophobic polymer matrix composites [8]. Moreover, NLFs are characterized by an intricate structure with variable cross-section dimensions within fibers of the same species [8,9,10], associated with different values of density and tensile strength. As such, investigation of the variation in these properties with the diameter of any NLF might be of utmost importance, aiming for possible industrial applications. The application of natural fibers as an alternative to synthetic fibers in polymer matrix composites already occurs in products available in the global market [11,12,13,14].

Brazil is a country that stands out in terms of native NLFs, in which there is a variety and availability of several fibers, with a potential application in engineering composites, but that have not yet been fully studied [15]. Moreover, the regional development of rural and riverside regions associated with the plantation of these fibers is another possible scenario for the employment of people involved in the production of NLFs. Thus, an extensive investigation on the use of polymer matrix composites reinforced with NLFs for ballistic protection confirms such employability [16,17,18,19].

Given this scenario, the study of fibers taken from a plant known in Brazil as “seven-islands-sedge” (*Cyperus malaccensis*), owing to its origin in the seven-island region in Japan. Worth mentioning is the absence of nodes on its sharp, long, triangular stem, with 2 to 3 leaves at the tip of the stem. The plant was cultivated in the Vale do Ribeira region in the state of São Paulo, Brazil [20,21], by Japanese immigrants.

As for any NLF, the arrangement of each fiber is influenced by the factors that control the process and growth of the complex structure controlled by the metabolism of the plant cells. These factors are related to the amount of light in the environment, soil composition, water abundance, and even the genetic variation in each plant. Therefore, the mechanical and physical properties are distinct among fibers [22,23,24]. There is also a tendency to increase the mechanical properties of NLFs related to the inverse of their diameter [8]. This occurs due to the presence of lower levels of defects present in thinner fibers when compared to larger diameter fibers, which leads to the thinner fibers acquiring better mechanical properties [25,26].

In this context, the present work aims to investigate some basic properties of the sedge fibers in terms of their density and frequency distribution of fiber diameter dimensions as well as the thermal analysis intrinsic to these fibers.

## 2. Materials and Methods

The seven-islands-sedge fibers, sedge fibers for short, investigated in this work were extracted from a mat purchased from the company Artevale, Brazil. A bundle of sedge fibers is shown in Figure 1. They were cleaned, immersed in water for 24 h, shredded and cut into 150 mm lengths, and then dried in an oven at 70 °C for 24 h. In addition, the fibers had not undergone any chemical treatment.

The use of the optical microscope, model BX53M, OLYMPUS, with a 5× magnification in the dark field mode served as an aid to measure the diameter of the fibers. A total of 100 samples were selected randomly from the bundle of fibers, after obtaining the fibers by the shredding process. The measurements were made by finding 5 regions at (0°) and 5 regions at (90°), distributed along the length of the samples; in each region, 3 measurements were made and an average was obtained. After this procedure, an average of the values obtained was again performed, in micrometers (µm), along with the fiber length. In this manner, it was possible to obtain the average of the characteristic diameters of each fiber.

Due to the extensive variety of fiber diameters, a frequency distribution was performed by considering 10 diameter intervals. This allows to create a histogram. In addition, the fibers had their mass measured by an electronic (0.0001 g) precision scale model FA2104N, Bioprecisa. It was then possible to preliminarily evaluate the density by measuring the geometric linear density.

The determination of the moisture content followed the ASTM D1348 [27]. The method is based on weighting the samples and immediately inserting them in an oven at a temperature of 105 °C, for a period of 2 h, to remove the moisture. After this period of time, the samples were weighted several times and taken to the oven for 30 min, until the weight loss, in successive weightings, acquired a difference of less than 0.005 g. A total of five samples were prepared and measured in the same abovementioned scale, as required by ASTM D1348 [27]. The moisture content was calculated according to Equation (1), where the $O_{S}$ is related to the sample containing moisture and *O_BD_* for a dried sample.
(1)$$M_{CBD}=\frac{O_{S}-\left(O_{BD}\right)}{O_{BD}}\times 100$$

In order to determine the true volume of a solid, a gas pycnometer analysis was carried out. For this, the fibers were comminuted and inserted into a chamber, then subjected to a degassing process in repeated purges with gas helium to remove the impurities and moisture. After the purges, the entire system is brought to atmospheric pressure, isolating the expansion chamber, closing the expansion valve, and pressurizing the chamber containing the sample inside to a pressure P1 (about 17 psi or 1.16 atm). The analysis was carried out following the procedures established by ASTM D4892 [28] and D550 standard [29].

For the thermogravimetry analysis (TGA) and differential scanning calorimetry (DSC), the fibers were comminuted and placed in the platinum crucible of the Shimadzu equipment, model DTG-60H. The sample was subjected to a heating rate of 10°/min, starting at 30 up to 600 °C under a nitrogen atmosphere. The DSC samples were placed in an aluminum crucible DSC-60 model of Shimadzu, subjected to a heating rate of 10°/min, starting at 30 °C up to 250 °C on a nitrogen atmosphere and with a gas flow of 50 mL/min. Morphological analyses of the surface of the fibers were performed by scanning electron microscopy (SEM) using a model Quanta FEG 250 Fei microscope operating with secondary electrons between 5 and 15 kV.

Values acquired for the density by diameter interval were statistically interpreted by means of Weibull Analysis software; the densities of the one hundred fibers were divided into 6 different intervals, considering an average diameter for each interval. The software provided the corresponding scale parameter (θ), the shape parameter (β), and the precision adjustment (R^2^) parameters. This is based on the cumulative Weibull frequency distribution function (F(x)), given by Equation (2), as a linear relationship in which β is the slope.
(2)$$ln\;ln\;\left[ln\;ln\;\left(\frac{1}{1-F\left(x\right)}\right)\;\right]=\beta \;ln\;ln\;\left(x\right)-\left[\beta \;ln\;ln\;\left(\theta \right)\;\right]$$

## 3. Results and Discussion

### 3.1. Density Variation with Fiber Diameter

Due to the fact that natural fibers present a significant heterogeneity related to their diameters, the objective was to investigate which frequencies of the diameter intervals are greater. Additionally, this frequency distribution of the sedge fiber diameter is presented in Table 1.

A low percentage of fibers is present in the intervals between 0.16 and 0.32 mm, which is possibly associated with the difficulty in obtaining fine fibers manually without causing fiber breakage. On the other hand, fibers found between 0.69 and 0.92 mm showed a low frequency, since the existence of a higher number of defects in thick fibers, making it difficult to obtain non-broken fibers at these frequencies. The frequency that presented the highest number of fibers was found between the 0.47 and 0.54 intervals. The same behavior has been observed in previous studies [30,31].

### 3.2. Weibull Analysis

The average fiber density was measured and statistically analyzed by the Weibull method. In addition, density measurements for 100 sedge fibers with the distinct cross-sectional area were performed by mass/volume and gas pycnometer analysis. The average density found for all one hundred fibers was 0.46 g/cm^3^ and the absolute density found by the gas pycnometer technique was 1.8861 g/cm^3^. The parameters, provided by the Weibull analysis software, are presented in Table 2.

The values of the parameter θ follow the same trend as the average density of the sedge fibers, where the value increases as the diameter decrease. The parameter R^2^ indicates the quality of the adjustment of the Weibull line created from the collected data. Values near or above 0.9 for R^2^ indicate good quality for the linear fit, revealing that the data are distributed according to the Weibull function. The parameter θ indicates the central value of the characteristic distribution. The parameter β is a measure of reliability, since the higher its value, the narrower the distribution. The highest values of β acquired (7.30 and 5.72) highlight the homogeneity of the sedge as a function of density and diameter [32]. The θ values followed the same trend as the average diameter values found for the fiber, in addition to the R^2^ parameter indicating a value closer to one for the average diameters of 0.61 and 0.86 mm, indicating a quality of precision adjustment, justifying the good approximation of the characteristic density associated with the average, providing good reliability to the statistical values. From the data presented, it was possible to create a graph of the characteristic density associated with an average diameter presented in Figure 2.

A trend of density decrease with an increase in diameter is observed, in agreement with what occurs for other FNLs present in the literature [8,33]. In order to observe how the characteristic density behaves associated with the diameter, the Weibull characteristic density values were related to the average diameters of each interval.

According to the literature, some authors related the properties of certain lignocellulosic fibers to the inverse of their diameter by a linear adjustment [34]. The inverse of the diameter associated with the characteristic density of the sedge fiber together with the linear approximation of its points is shown in Figure 3.

Based on the linear approximation, it was possible to create a mathematical correlation, and a hyperbolic equation was proposed to fit the data in Equation (3).
(3)$$\rho =\frac{0.16807}{D}+0.16291$$

### 3.3. Scanning Electron Microscopy (SEM)

The correlation allows us to observe that the density values tend to approach the value of 0.16291 g/cm^3^ as the diameter values increase. On the other hand, density tends to increase when the diameter values become smaller. An increase in density, which is directly related to the decrease in diameter, can be attributed to fewer defects in the finer fibers [32]. The fourth and fifth intervals do not follow the linear trend of the others, due to the small variation between the existing values in the diameter intervals. Actually, both intervals have low values for the parameters of R^2^ and β responsible for indicating the quality of the fit associated with a linear approximation. The SEM micrographs in Figure 4 demonstrate more clearly the difference between the surfaces of the thinner and thicker sedge fibers.

The presence of a wavy surface in Figure 4a and the presence of voids inside the fiber can act as a mechanism that indicates the reason why thick fiber presents a low-density value. Figure 4b shows a more homogeneous surface with no surface defects corroborating with the parameters that provide an improvement in mechanical strength as reported in the literature [35].

### 3.4. Moisture Content

The moisture data obtained are represented in Table 3, showing the mass acquired until the loss of mass was equivalent to 0.005 g, it being necessary to repeat the process four times.

The percentage of moisture content found in the sedge fibers was 15.2%, this being higher than the other values presented by other fibers already researched, such as coir (10%), flax (7%), hemp (9%), and sisal (11%) [36]. Therefore, the drying process is necessary to ensure good adhesion of the matrix interface and the reinforcement interface [35,37].

### 3.5. Thermogravimetric Analysis (TGA)

Figure 5 shows the results of TGA and first-order derivative analysis (DTG). A mass loss of 10.2% can be observed between 26 and 100 °C, due to evaporation of the moisture content, causing dehydration of the fibers. Good thermal stability is also observed up to the temperature of 241 °C at which the mass loss was 13.23%.

A sudden mass drop is observed from 241 to 351 °C, showing a mass loss of 63.07%. In the temperature range from 351 to 496 °C, the fiber had a total mass loss, in total 96.04%. After this range, no further degradation and mass loss occurred until the end of the test at 600 °C, due to the ash content present.

The temperatures of 266.68 °C (T_onset_) and the maximum rate at 300.95 °C are related to the onset of degradation of the structural components of the sedge fiber, such as lignin and hemicellulose. While, around 349 °C, another mass loss is associated with the onset of cellulose degradation, as mentioned in the literature [38].

### 3.6. Differential Scanning Calorimetry (DSC) Results

Thermogravimetric analysis of the sedge fiber demonstrated higher thermal stability than fibers of the same botanical genus, such as the fiber known as *Cyperus pangorei*, where Mayandi et al. [39] observed thermal stability up to 221 °C, adopting the same parameters related to nitrogen atmosphere with a heat rate of 10 °C/min. Furthermore, the final mass loss phase occurred at 324 °C, which corresponds to the degradation of alpha-cellulose and lignin [40,41]. Figure 6 shows the DSC curve of the sedge fiber.

At 97.4 °C, an endothermic peak associated with moisture loss, due to heating, is observed by the sedge fiber [42]. The increase in heat flux around 250 °C is associated with the presence of an exothermic peak responsible for a higher mass loss from 255 °C onwards, corroborating the results of the TGA analysis.

## 4. Summary and Conclusions

A Weibull statistical analysis of the measured density for manually cut and unraveled sedge fibers presented an inverse correlation between the density and the fiber mean diameters. Good R^2^ quality was found at the four and six mean diameter intervals, which indicates that data are distributed according to the Weibull function.

The absolute density was performed by a gas pycnometer analysis and it was considerably higher (1.8861 g/cm^3^) when compared to the density obtained by the linear geometric method. The moisture content present in sedge fibers is higher than in other fibers investigated in the literature.

SEM analysis provided evidence that a thicker sedge could have a lower density since they present more hollow spaces inside of them that act as a mechanism of rupture.

TG/DTG parameters for the sedge revealed that thermal stability is up to 241 °C and an intense degradation peak at 300.95 °C.

DSC analysis of the fibers showed an endothermic peak at 97.4 °C, due to moisture evaporation; also, an increase in the heat flow around 250 °C is possibly associated with an exothermic peak presence responsible for generating a higher loss of mass, as was observed with the same pattern on TGA analysis.

In terms of engineering applications, the investigated material has great potential in the manufacture of composites. Characterization of the reinforcement of polymeric matrices and certain treatments can be applied to these fibers to improve the properties of the composite and to provide accurate knowledge of which engineering or ballistic applications they can be subjected to.

## Figures and Tables

**Figure 1 polymers-14-03687-f001:**
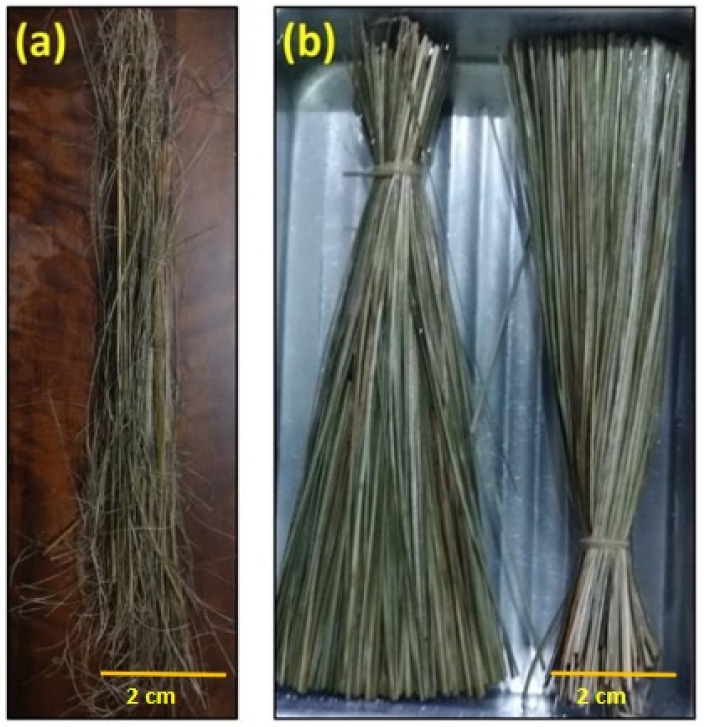
(**a**) Sedge fibers as received; (**b**) sedge fibers immersed in water for 24 h.

**Figure 2 polymers-14-03687-f002:**
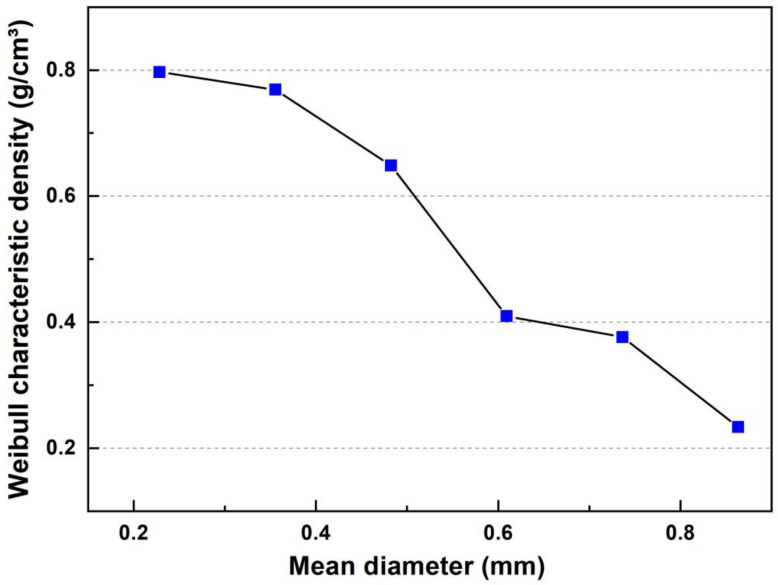
Weibull characteristic density of the sedge plot associated with their mean diameter.

**Figure 3 polymers-14-03687-f003:**
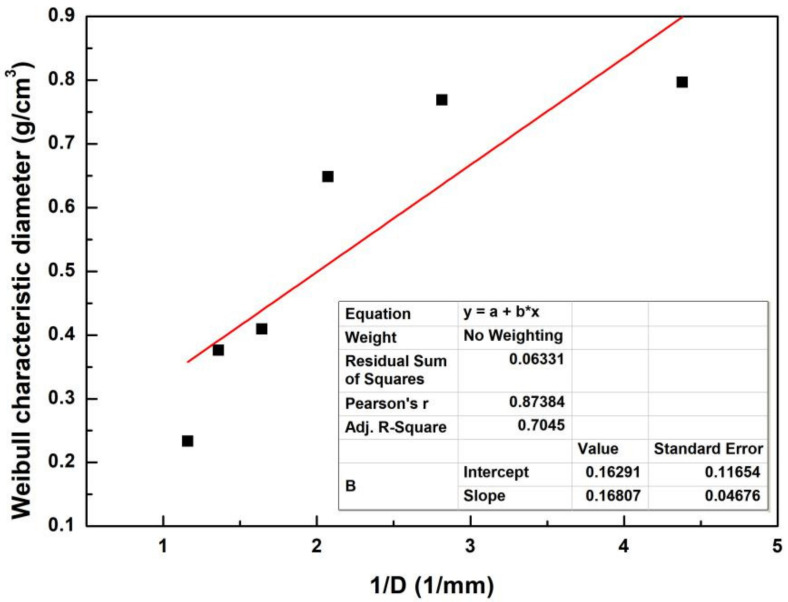
Characteristic density plot related to the inverse of the mean diameter.

**Figure 4 polymers-14-03687-f004:**
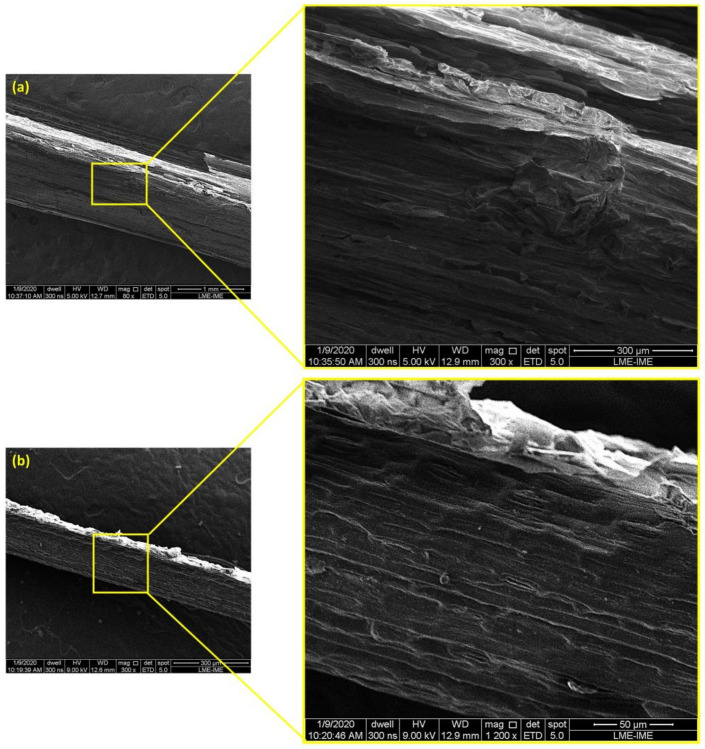
SEM images of the surface of the sedge fibers: (**a**) Surface of the thicker fibers exhibiting higher density of defects—magnifications of 80 and 300×; (**b**) Surface of the thinner fibers showing fewer defects along the fiber—magnifications of 300 and 1200×.

**Figure 5 polymers-14-03687-f005:**
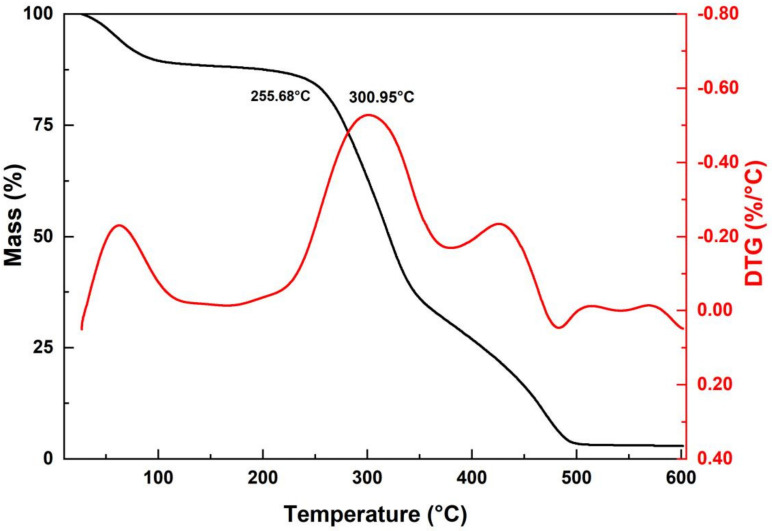
TGA/DTG curve of the sedge fibers.

**Figure 6 polymers-14-03687-f006:**
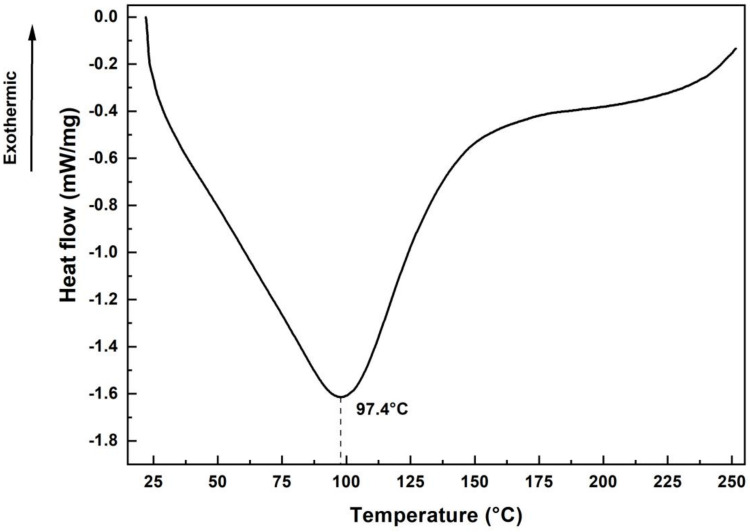
DSC curve of the sedge fibers.

**Table 1 polymers-14-03687-t001:** Frequency distribution of the sedge fiber diameter.

Diameter Intervals	Intervals (mm)	Frequency (%)
1	0.16–0.24	1
2	0.24–0.32	2
3	0.32–0.39	21
4	0.39–0.47	17
5	0.47–0.54	31
6	0.54–0.62	14
7	0.62–0.69	10
8	0.69–0.77	1
9	0.77–0.85	2
10	0.85–0.92	1

**Table 2 polymers-14-03687-t002:** Characteristic mean diameter for each interval of sedge fiber diameter.

Mean Diameter (mm)	Mean Density (g/cm^3^)	Statistical Deviation (g/cm^3^)	Weibull Modulus (β)	Characteristic Density (θ) (g/cm^3^)	Precision Adjustment (R^2^)
0.23	0.74	0.14	5.72	0.79	0.84
0.36	0.68	0.26	2.87	0.77	0.79
0.48	0.61	0.35	1.69	0.65	0.64
0.61	0.38	0.08	5.34	0.41	0.93
0.74	0.34	0.11	3.36	0.38	0.69
0.86	0.22	0.03	7.30	0.23	0.96

**Table 3 polymers-14-03687-t003:** Moisture content percentage of sedge fibers samples.

Samples	Mass (g)	2 h	2 h 30 min	3 h	3 h 30 min	Moisture Content (%)
1	0.43	0.38	0.38	0.35	0.38	13.57
2	0.63	0.57	0.56	0.55	0.55	13.26
3	0.72	0.62	0.61	0.61	0.62	15.70
4	1.21	1.09	1.07	1.07	1.07	13.46
5	0.49	0.43	0.42	0.42	0.42	16.96

## Data Availability

The data presented in this study are available on request from the corresponding author.

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
