# Peer review of "Evaluation of the Change in Density with the Diameter and Thermal Analysis of the Seven-Islands-Sedge Fiber"

_polymers, 2022, doi:10.3390/polym14173687_

Round 1

Reviewer 1 Report

Manuscript No. polymers-1866562

"Evaluation of the change in density with the diameter and thermal analysis of the seven-islands-sedge fiber"

Comments:

--Are there any differences on structure and properties of fibres grown at seven islands? 

--The estimation of fibre density using the proposed Equation (3) is not encouraging. 

The linear fitting result with R^2 around 0.7 cannot lead to a linear approximation. The inverse relationship between fibre diameter and density proposed in this study was derived from Weibull analysis.  However, the analysis was used largely in this field to evaluate natural fibre shape variation effects on fibre mechanical properties (see e.g. Shuttleworth et al, Cellulose. 2019;26(8):4693-4706.  Boubakar et al, J Mater Sci. 2017;52(11):6591-6610. ), which has to be discussed here or include as background info. in Introduction. 

So, additional experiments on fibre mechanical properties and structure-strength relationship are needed to address these concerns.

--Thermal analysis very confusing. Many mismatches are easily found in the text. Examples (not limited to):

- 'A mass loss of 10.2% can be observed between 26 and 100°C' however, Fig.7 shows <5% mass loss at 100 °C.

-These two temperatures (300, 334 °C) are highghted in Fig. 7, but not explained.

-These two temperatures (66.8, 122.9 °C) are highghted in Fig. 8, yet authors discussed thermal events at totally different temperature range. 

--Graphic presentation needs to be largely improved. 

-Fig.1, it might be good to include fibre treatment details i.e., cutting, washing, drying, and measuring; Scale bar is also suggested. 

-Fig.2 and Table 1 basically show the same information on fibre diameter distribution, either one can be deleted.

-Fig.3 It's really difficult to read anything from subfigues.

-Fig.5 For fitting purpose, dotted data rather than dot+line are allowed. Besides, author can explain the fitting result in maintext, not Inset table that is hard to see.

-Fig.7 Y-axis must be Mass, not mass loss.

-Fig.8 Arrows to indicate endothermic/exothermic peaks

Author Response

All the revisions have been done.

Reviewer 2 Report

Dear Authors

The article focus on a good topic specially using weibull  distribution.  However Figure 3 is hardly readable. It need to be improve. Hardly X axix , y axix, inset data, is readable. This is the important part of the analysis, however its not readable. Figure 5 inset data from weibull has similar problem, author need to improve this.  Fig. 8 DSC curve  should have a base line, , what the peak signifies?

Although author mention  about 97.4, harrdly it marked in DSC graph.  In conclusion author should specify the findings from Weibull. 

Author Response

All the revisions have been done.

Reviewer 3 Report

Dear Authors,
I found your manuscript of high interest to readers due to the valuable and up-to-date topic. However, below, please find several remarks, which in my opinion help to make your manuscript better:
- line 83 and 85, where you describe the moisture content measurement, please say "weighting" instead of measuring (line 83) and "weighted" instead of measured (line 85)
- line 89 - remove "content" from "containing moisture content"
- line 121 (Figure 2) 1) please use a dot instead of a comma in decimals (in the entire manuscript, also on figures and tables, 2) since you presented the same data in Figure 2 and Table 1, something is redundant; I suggest to remove Figure 2
- line 135 - 137 - these two sentences should be moved to the Methodology section
- line 145 (Figure 3) - the plots are extremely hard to read; please try to enlarge the fonts
- line 204 - 205 - the first sentence should be moved to the Methodology section
- line 209 (Table 3): 1) Why does the mass of samples no. 1 and 3 rise in 3h 30min referred to 3h? 2) Please keep the same precision (two numbers after the dot) in the entire table (see sample no. 5 mass in 2h)
- line 211 -213 - your comment about the moisture content of the fibers seems to be unuseful, when not referred to the storage conditions of the fibers before moisture content measurement. For the future, I suggest storing the investigated material in a known environment (for example 20 deg. C and 65% humidity) to mass stabilization, and then measure the moisture content
- line 218 - the information about heat rate and the atmosphere was given in the Methodology section, so here can be removed
- line 223 (Figure 7) - worth continuing the TGA measurement since the plot still goes down under the temp. of 600 deg. C, that means the mass is still decreasing, to see what was the ash content
- line 227 - 228 - the sentence "After this range... [...] ...ash content present." in my opinion is not supported by the data achieved, because, as was shown in Figure 7, the mass was still decreasing at 600 deg. C and the TGA test should be continued to higher temperatures
- line 238 - 240 - how can you comment that "the final mass loss phase occurred at 324 deg. C", since, according to Figure 8, which is commented this way, the maximum temperature you reach is 275 deg. C only?

Best regards!

Author Response

All the revisions have been done.

Round 2

Reviewer 1 Report

The manuscript has improved after major revision.

One more comment:

Summary and Conclusion. Authors have only summarised instrumental testing results. Yet no any instructive and concluding remarks mentioned in this section.

Author Response

All revisions have been done.

Reviewer 2 Report

Authors significantly improved the article. Accept. 

Author Response

All revisions have been done.
